# The Morphological and Molecular Characterization of the Avian Trematodes *Harrahium obscurum* and *Morishitium dollfusi* (Digenea: Cyclocoelidae) from the Middle Volga Region (European Russia)

**DOI:** 10.3390/biology13080621

**Published:** 2024-08-15

**Authors:** Alexander A. Kirillov, Nadezhda Yu. Kirillova, Sergei V. Shchenkov, Alexei E. Knyazev, Victoria A. Vekhnik

**Affiliations:** 1Laboratory for Zoology and Parasitology, Institute of Ecology of Volga River Basin RAS, Samara Federal Research Scientific Center RAS, Togliatti 445003, Russia; nadinkirillova2011@yandex.ru (N.Y.K.); lexkame@gmail.com (A.E.K.); ivavika@rambler.ru (V.A.V.); 2Department of Invertebrate Zoology, Saint Petersburg State University, St. Petersburg 199034, Russia; svshchenkov@yandex.ru

**Keywords:** trematoda, cyclocoelids, Aves, Volga region, phylogenetics

## Abstract

**Simple Summary:**

Parasitic flatworms of the family Cyclocoelidae parasitize various inner organs of birds, mainly in their respiratory and digestive systems. The taxonomic position of many cyclocoelid species remains controversial. In our study, we studied two species of trematodes from the family Cyclocoelidae (*Harrahium obscurum* and *Morishitium dollfusi*) using classical morphological and molecular phylogenetic approaches. We provided original drawings, morphological descriptions, and the results of a molecular phylogenetic analysis of these species of helminths from birds in Russia for the first time. Our work shows the morphological variability in individuals of both flatworm species. Since the variability of individual cyclocoelids may be rather high, morphological features alone are insufficient for their identification, and confirmation with the use of molecular data is required.

**Abstract:**

The taxonomic status of many species of the family Cyclocoelidae is still unclear. Two species of cyclocoelids, *Harrahium obscurum* and *Morishitium dollfusi*, were collected from the air sacs of birds (*Turdus merula* and *Tringa ochropus*) inhabiting the Middle Volga region (European Russia). Here, we provide the first detailed morphological description of these cyclocoelids and combine it with the first molecular phylogenetic analysis of Cyclocoelidae from birds in Russia based on partial sequences of their 28s rDNA and coI mtDNA genes. Specimens of both flatworm species from different host individuals differ slightly in body shape and size, which probably reflects host-induced intraspecific variability. For the first time, we have shown that a stable morphological character such as the length of the vitelline fields in the studied digeneans is variable at the species level and cannot be used in their morphological diagnosis.

## 1. Introduction

The family Cyclocoelidae was erected more than 120 years ago. Despite this, the taxonomic status of many cyclocoelid species is still unclear [1]. The morphological differences between some genera of the family Cyclocoelidae are insignificant, which complicates their species identification [2]. There are also very limited molecular data on cyclocoelids of the world’s fauna [1,3,4,5,6,7].

Trematodes of the family Cyclocoelidae are represented in Russia by 11 genera and 29 species. Of these, 18 species from 9 genera are known parasites of birds in European Russia [2,8,9,10,11]. However, there are no molecular genetic data on cyclocoelids in Russia.

The cosmopolitan species *Harrahium obscurum* (Leidy, 1887) (formerly *Cyclocoelum obscurum*) parasitizes the air sacs of various birds of the order Charadriiformes [1]. In Russia, this cyclocoelid species was identified only from *Gallinago gallinago* (Linnaeus, 1758), *Numenius arquata* (Linnaeus, 1758), *Limosa limosa* (Linnaeus, 1758), and *Tringa totanus* (Linnaeus, 1758) in the territory of Dagestan [9]. 

According to Sitko et al. [1], *Harrahium* spp. usually parasitizes waders of the family Scolopacidae, while *Cyclocoelum mutabile* (Zeder, 1800) usually parasitizes Rallidae birds. Therefore, we believe that *H. obscurum* could previously have been identified as *C. mutabile* or *Cyclocoelum pseudomicrostomum* Harrah, 1922 in Russian waders. Additionally, we know of similar are known for shorebirds of the Komi Republic, Leningrad, and Rostov regions, the White Sea, the Middle Volga region, the Volga delta, the Urals, Western and Eastern Siberia, and Primorsky Kraj [8,11,12,13,14,15,16,17,18]. In the Middle Volga region, *Cyclocoelum* trematodes (as *C. mutabile*) were found in *G. gallinago*, *L. limosa,* and *Vanellus vanellus* (Linnaeus, 1758) in Bashkortostan, as well as in *Tringa ochropus* Linnaeus, 1758 in the Nizhny Novgorod region [13,17]. Unfortunately, these works do not present the morphological and morphometric characteristics of the studied cyclocoelids, nor any molecular data.

*Morishitium polonicum* (Machalska, 1980) was previously considered to be a specific parasite of the air sacs of thrushes in Europe [1,6,19,20] and was first recorded by us in Russia in the common blackbird *Turdus merula* Linnaeus, 1758 and in the hawfinch *Coccothraustes coccothraustes* (Linnaeus, 1758) [10]. According to a recent study by Heneberg and Sitko [6], *M. polonicum* is a junior synonym of *Morishitium dollfusi* (Timon-David, 1950), a common parasite of European corvids and thrushes.

In this study, we collected and analyzed the morphological and morphometric characteristics of two cyclocoelid species and for the first time presented molecular phylogenetic data on cyclocoelids from Russia.

## 2. Materials and Methods

### 2.1. Trematodes’ Collection and Morphological Examination

During a comprehensive parasitological survey of birds in the Middle Volga region (European Russia), the helminth fauna of 8 individuals of *Tringa ochropus* and 11 individuals of *Turdus merula* was studied in April and August–September of 2022 and 2023. Adult specimens of cyclocoelids were collected from birds in two locations in the Middle Volga region: the Samara region and the Republic of Mordovia. 

Cyclocoelid trematodes were recovered from the air sacs of three *T. ochropus* and two *T. merula* and preliminarily identified as *H. obscurum* and *M. dollfusi*, respectively. In total, 14 specimens of *H. obscurum* and 136 specimens of *M. dollfusi* were collected. Only alive motile digeneans were selected for the study. For morphological examination, 20 specimens of *M. dollfusi* and 12 specimens of *H. obscurum* were immobilized by heating in saline. The parasites were then stained with acetic carmine, dehydrated, cleared with clove oil, and mounted in Canada balsam. Specimens of cyclocoelids from each host species were fixed in 96% ethanol and stored at +4 °C for molecular phylogenetic analysis. The drawings were made using an MBI-9 light microscope (Lomo, Saint Petersburg, Russia) with a Levenhuk M500 BASE Digital Camera (Levenhuk, Inc., Tampa, FL, USA) and a RA-7 drawing tube. All measurements are given in millimeters. Morphological identification was carried out using the keys of Bashkirova [21], Dronen and Tkach [22], and Dronen and Blend [2]. Voucher specimens were deposited at the Institute of Ecology of the Volga River Basin of the Russian Academy of Sciences (Togliatti).

### 2.2. Molecular Data and Phylogenetic Analysis

Total DNA was isolated from small tissue sections of individual specimens using Chelex-100 (Bio-Rad, Hercules, USA) with a Proteinase-K solution (diaGene, Moscow, Russia). The primers ‘28sy/28sz’ [23] and ‘JB3/JB4.5’ [24] were used to amplify partial 28S rDNA and coI mtDNA, respectively. PCR parameters were set according to Kirillova et al. [25]. Amplicons were sequenced at “Evrogen” (Moscow, Russia).

To assess the phylogenetic position of the species, the 28S rDNA gene dataset (Table 1) was analyzed according to a pipeline implemented in the “R” programming language [26]. The code (including the Bayes block) is available at the following GitHub repository: ‘https://github.com/Shchenkov/dirty_seqs’ (accessed on 22 May 2024). A Bayesian Inference analysis was performed with Mr. Bayes 3.2.7a on a local workstation. The quality of the chains was estimated using the built-in MrBayes tools. Based on estimates, the first 25,000 generations were discarded for burn-in. The final length of the alignment was 1043 bp. We included *Himasthla elongata* (Mehlis, 1831) as an outgroup in our analysis following the results of Perez-Ponce de Leon and Hernandez-Mena [27]. Pairwise distances were calculated from coI mtDNA sequences in MEGA 11 software using standard parameters [28]. A heatmap based on pairwise distances was made using the “ComplexHeatmap” library [29] in the “R” programming language.

## 3. Results

### 3.1. Systematic and Morphological Characteristics of the Studied Cyclocoelids

Family Cyclocoelidae Stossich, 1902

Genus *Harrahium* Witenberg, 1926

*Harrahium obscurum* (Leidy, 1887) (Figure 1)

Host: the green sandpiper *Tringa ochropus* Linnaeus, 1758 (Scolopacidae).

Site of infection: air sacs of lungs.

Locality: Mordovia Nature Reserve (Republic of Mordovia, 54.932270° N, 43.421529° E; 54.726151° N, 43.148885° E; 54.744993° N, 43.116004° E).

Prevalence: 37.5% (3 of 8 birds).

Intensity: 2–9 per infected wader.

Mean intensity: 1.75.

Accession numbers in collection of IEVB RAS: No 2301–2306.

Availability (Representative DNA sequence): GenBank No 28S rDNA—PP935228, coI mtDNA—PP934665).

General description of *Harrahium obscurum* (based on 12 mature specimens): Body lanceolate, elongated, tapered anteriorly and rounded posteriorly. Mouth subterminal. Oral sucker vestigial. Ventral sucker absent. Pharynx small, oval. Esophagus thin, more or less S-shaped. Ceca extend to body end, united behind the posterior testis and forming cyclocoel. Testes, oval, entire, lie in posterior body part. Posterior testis located medially and lies directly in cyclocoel. Anterior testis lies at some distance from posterior one and adjacent to cecal branch. Cirrus sac pear-shaped, located at level of esophagus, and does not reach level of cecal bifurcation. Genital pore medial, located at approximately midlevel of pharynx. Ovary round, intertesticular, much smaller than testes, forms a triangle with testes and lies on side opposite anterior testis. Ovary separated from posterior testis by Mehlis’ gland. Mehlis’ gland (complex) round, located dorsally posterior to ovary, approximately the same size as ovary. Uterine seminal receptacle small, pear-shaped. Laurer’s canal not observed. Vitellarium consists of small follicles of irregular shape. Vitelline fields extend along ceca laterally from level of ceca bifurcation (or slightly above) to posterior extremity, not confluent posteriorly. Transverse vitelline ducts run along anterior margin of posterior testis. They unite behind the Mehlis’ complex and form the vitelline reservoir. Uterus intercecal, fills entire space between ceca branches and uterus loops does not extend beyond them. Metraterm well developed.

Remarks: All specimens of *H. obscurum* from different individuals of *T. ochropus* were morphologically similar. At the same time, variability in body and organ sizes was observed among different individual trematodes. The morphometric characteristics of the studied specimens of *H. obscurum* are presented in Table 2.

Genus *Morishitium* Witenberg, 1928

*Morishitium dollfusi* (Timon-David, 1950) (Figure 2)

Syn.: *Morishitium polonicum* (Machalska, 1980) 

Host: the common blackbird *Turdus merula* Linnaeus, 1758 (Turdidae).

Site of infection: air sacs of lungs.

Locality: vicinity of Kashpir village (Samara region, 53.004919°N, 48.554057°E).

Prevalence: 18.2% (2 of 11 birds).

Intensity: 56–70 per infected bird.

Mean intensity: 12.4.

Accession numbers in collection of IEVB RAS: No 2307–2316.

Availability (Representative DNA sequence): GenBank No 28S rDNA—PP935227, coI mtDNA—PP934664.

General description of *Morishitium dollfusi* (based on 20 mature specimens): Body lanceolate, elongated, tapered anteriorly and rounded posteriorly. Oral sucker vestigial. Ventral sucker absent. Pharynx round. Prepharynx short. Esophagus thin. Ceca extend along the body to posterior end, united behind posterior testis and forming cyclocoel. Testes oval, entire, located in the posterior body part. Posterior testis always larger than anterior one, located along the body midline and lies directly in cyclocoel. Anterior testis displaced from the midline to cecal wall, located at some distance from the posterior one. Cirrus sac pear-shaped, located at esophagus level. Cirrus sac base reaches cecal bifurcation, but does not extend below it. Genital pore postpharyngeal, medial or slightly submedial. Ovary round and lies between testes, approximately in line with them. Mehlis’ gland approximately the same size as ovary and located dorsally, at posterolateral margin of ovary. Uterine seminal receptacle small, pear-shaped. Laurer’s canal not observed. Vitellarium consists of small follicles of irregular shapes, begins approximately at the level of the posterior margin of cecal bifurcation and does not extend beyond bifurcation. Vitelline fields extend laterally to posterior body end, not confluent posteriorly. Transverse vitelline ducts lie at level of anterior margin of posterior testis, unite behind Mehlis’ gland, and form vitelline reservoir. Uterus intercecal, occupies entire space between ceca branches and does not extend beyond them. Metraterm well developed.

Remarks: All specimens of *M. dollfusi* from both infected individuals of *T. merula* were morphologically similar (Figure 2A–C). At the same time, in different specimens of trematodes, variability was observed in the size of the body and inner organs, as well as in the position of the vitelline fields (Figure 2, Table 3).

### 3.2. Molecular Phylogenetic Analysis

Newly obtained sequences of the 28S rRNA gene of *M. dollfusi* were clustered with previously obtained sequences of the same species (Figure 3). *Morishitium polonicum* was found to be the sister clade of *M. dollfusi,* with full Bayesian support. *Circumvitellatrema nomota* was the sister clade of *M. dollfusi* and *M. polonicum*. The sequence of *H. obscurum* obtained in this study clustered with the sequence of *H. obscurum* previously described as *Cyclocoelum mutabile* (AY222249). Together, these sequences clustered with *Neohaematotrephus arayae* and then with *Anativermis normdroneni*. The sequences of *Tracheophilus cymbius* and *Typhlocoelum* sp. were clustered, with full Bayesian support, and together occupied the basal position among all cyclocoelids.

The pairwise distances between the newly obtained partial coI mtDNA sequence of *M. dollfusi* and the previously obtained ones are equal to 0.01 (Figure 4). At the same time, previously published sequences are identical to each other. The coI mtDNA sequence of *H. obscurum* we obtained is identical to one previously published and is at a small phylogenetic distance (0.01) from other available sequences. *Harrahium obscurum* is at intergeneric phylogenetic distance from *M. dollfusi* specimen. The phylogenetic distances between other specimens studied correspond to the levels of their phylogenetic positions, i.e., they are predictably consistent with intraspecific, interspecific, and intrageneric levels.

## 4. Discussion

In this study, we analyzed the morphological features and morphometric characteristics of two cyclocoelid species—*H. obscurum* and *M. dollfusi*—from two bird species of the Middle Volga region (European Russia) and obtained new molecular phylogenetic data on these parasites. The combined use of morphological and molecular phylogenetic methods made it possible to reliably identify the two cyclocoelid species we studied.

In general, the results of a molecular phylogenetic analysis based on the currently available and newly obtained sequences of cyclocoelids are partially consistent with previously published data. According to the results of the p-distance estimation, the species identification was correct in both cases, although it is quite difficult to identify cyclocoelid species using only morphological criteria (see below). Our phylogenetic analysis of partial 28S rRNA gene sequences revealed that Hyptiasminae and Cyclocoelinae are polyphyletic subfamilies, whereas Haematotrephinae is paraphyletic clade.

The morphology of all specimens of *H. obscurum* and *M. dollfusi* is consistent with their previous descriptions [1,7,10,19,20,30,37,38,39]. The specimens of two cyclocoelid species in our study generally corresponded to their previously presented morphometric characteristics (Table 2 and Table 3). Some differences were revealed in the morphometric characteristics of the studied specimens of both cyclocoelid species. Thus, the body length and width and the size of the reproductive organs varied in individuals of both of the species examined. In addition, *M. dollfusi* specimens showed variability in their body shape, the length of their vitelline fields, and the position of their genital pore. Among *Morishitium* specimens with a normal body shape (Figure 2A,C), there were individuals with a narrower, elongated body (Figure 2B). The anterior margin of the vitelline fields on the left and right can be either at the same level or at different levels (Figure 2). There was also variability in the position of the genital pore, which was located both medially and submedially in different *Morishitium* specimens.

We have provided the first complete morphological description of *H. obscurum* from *T. ochropus*, including morphometric data. The *H. obscurum* specimens we studied differed from those previously described in the length of their esophagus and post-testicular space (Table 2). Our specimens of *Harrahium* had a shorter esophagus and their post-testicular distance was greater than in previously described *H. obscurum* specimens [37,38,39].

The location of the genital pore is used in the identification of the genera of Cyclocoelidae [2,41]. In all studied specimens of *H. obscurum*, the genital pore was located approximately at the midlevel of the pharynx or somewhat posteriorly. Sitko et al. [1] describe a prepharyngeal location of the genital pore in their *H. obscurum* ex *T. erythropus* specimens. At the same time, Lopez-Jimenez et al. [30] noted that a contraction of the body can lead to changes in the position of the genital pore. Their work revealed the variability in the location of the genital pore from the anterior margin of the pharynx to its midlevel. In addition, the vitelline fields in the trematode specimens we studied always began at the level of the cecal bifurcation or slightly higher. On the other hand, in specimens from the works of other authors, the vitellarium began at the level of the pharynx [1,37]. Therefore, we believe that these two morphological features cannot be used in the morphological diagnosis of *H. obscurum*.

Our specimens of *M. dollfusi* had a shorter esophagus than specimens from previous descriptions (Table 3). In all studied specimens of *M. dollfusi*, the vitelline fields did not extend anteriorly to the intestinal bifurcation (Figure 2). In contrast, in the previously described *M. dollfusi*, the vitelline fields, at least on one side, reached the level of the oral sucker [1,2]. Therefore, as in the case of *H. obscurum*, the extent of the vitelline fields cannot be used in the diagnosis of *M. dollfusi*. A number of studies have also questioned the usefulness of the anterior and/or posterior distribution/extent of the vitelline fields in cyclocoelids as a diagnostic characteristic [42,43,44,45,46].

Different species of Cyclocoelidae are often morphologically similar and exhibit intraspecific variability [2,7,42,43,44,45,46]. This similarity can lead to confusion during species identification. Our study showed that a relatively constant morphological feature in the studied specimens of both cyclocoelid species is the position of the gonads. Thus, in *H. obscurum,* the ovary forms a triangle with the testes, and the ovary is always separated from the posterior testis by Mehlis’ gland. In *M. polonicum,* the ovary lies approximately on the same line as the testes, and Mehlis’ gland is located on the posterolateral edge of the ovary. Therefore, we believe that this morphological characteristic (the position of the gonads relative to each other) retains its significance as a generic status feature in the morphological diagnosis of cyclocoelids. However, some studies note variability in the position of the ovary and testes in cyclocoelids [2,46,47]. Dronen and Blend [2] attribute this to the procedure used for sample preparation.

Of the 11 examined individuals of *T. merula* from the Samara region, only two adult thrushes were infected with *M. dollfusi*. The parasite was not detected in 9 young birds. Our previous studies also showed the absence of an infection of young thrushes with *Morishitium* trematodes in Mordovia [10,48]. This may indicate that the infection of birds with *M. dollfusi* occurs in the Mediterranean region, where the wintering grounds of thrushes are located [49]. In addition, the life cycle of *M. dollfusi* includes terrestrial snails (mainly of the genus *Helicella* Ferussac, 1821) as intermediate hosts [19,20,50,51]. Gastropods of this genus do not live in the Middle Volga region. In spring (mid-April), we also found *H. obscurum* trematodes in three adult waders of *T. ochropus*. Five young waders were not infected with trematodes, which also indicates that birds become infected with this parasite in their wintering areas (Southern Europe, Africa, and South Asia) [52].

## 5. Conclusions

We have provided the first detailed morphological and morphometric description of two cyclocoelid species, *Harrahium obscurum* and *Morishitium dollfusi*, and presented the first molecular data on these cyclocoelids from birds in Russia. Specimens of *H. obscurum* and *M. dollfusi* from different individuals of both host species differ slightly in their body shape and the size of their internal organs, which probably reflects host-induced intraspecific variability. Overall, our data complement and expand our knowledge on the distribution of these two cyclocoelids.

## Figures and Tables

**Figure 1 biology-13-00621-f001:**
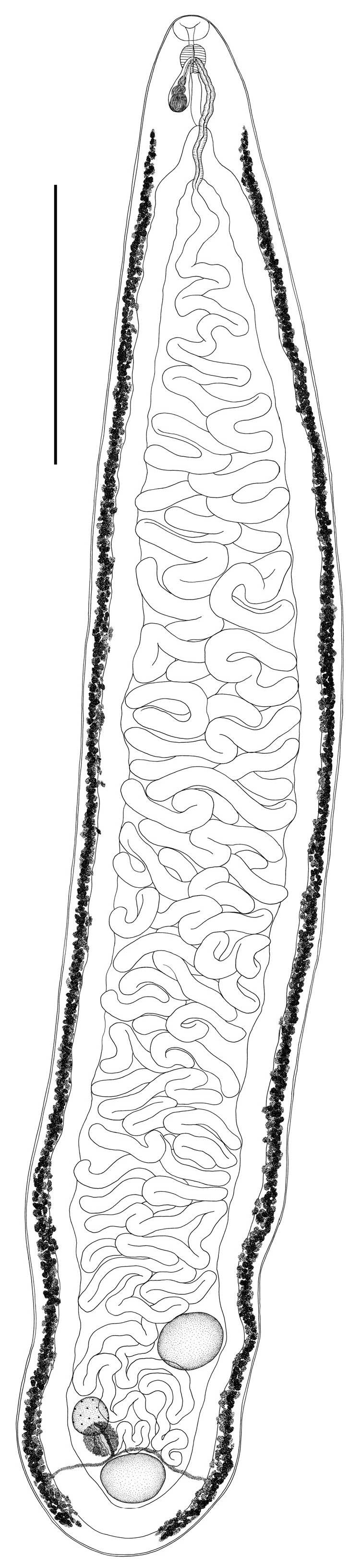
General morphology of *Harrahium obscurum* ex *Tringa ochropus*, ventral view; scale bar = 2.0 mm.

**Figure 2 biology-13-00621-f002:**
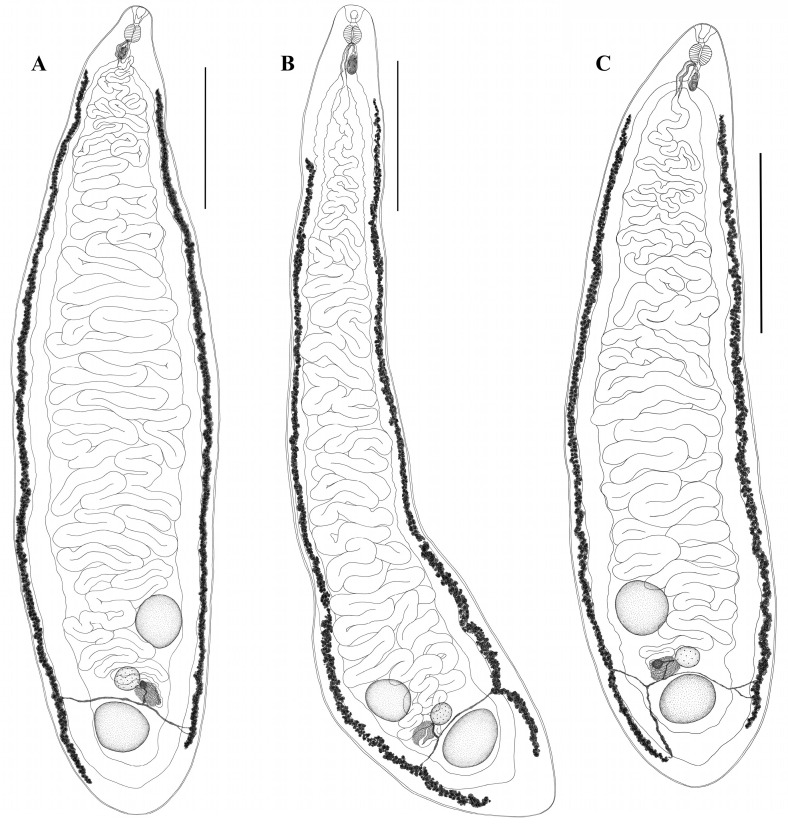
General morphology of *Morishitium dollfusi* ex *Turdus merula*, ventral view. (**A**–**C**)—morphological variations; scale bars = 2.0 mm.

**Figure 3 biology-13-00621-f003:**
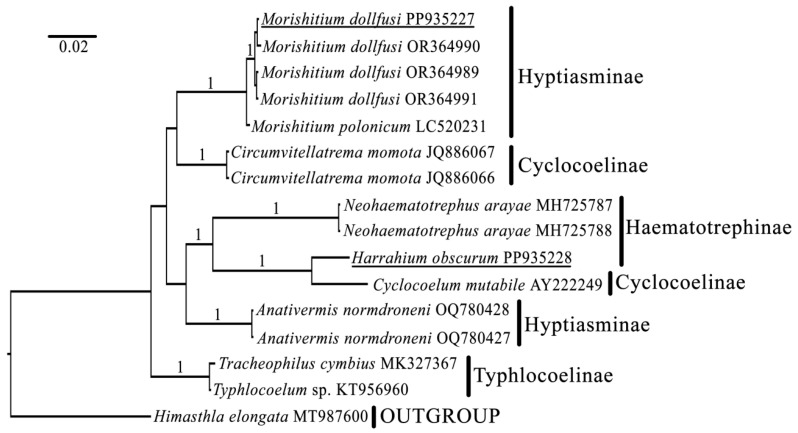
Phylogenetic position of *Harrahium obscurum* and *Morishitium dollfusi* based on partial 28S rRNA gene sequences. Only nodes with supports greater than 0.97 (BI) are shown. Newly obtained sequences are underlined.

**Figure 4 biology-13-00621-f004:**
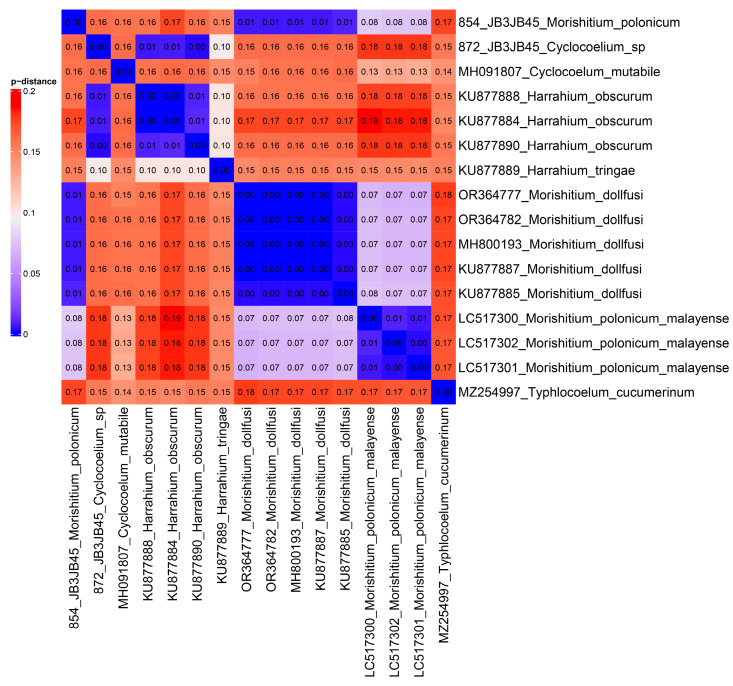
Heatmap based on the p-distances between partial coI mtDNA sequences. The values are shown in the corresponding cells. Pairwise distances are encoded into the color.

**Table 1 biology-13-00621-t001:** List of trematode specimens, with GenBank accession numbers, according to their geographic origin.

Family	Species	Host	Locality	GenBank Number		References
				28S rDNA	*cox1* mtDNA	
Cyclocoelidae	*Anativermis normdroneni*	*Branta canadensis* (Anatidae)	USA	OQ780427	–	[30]
	*Anativermis normdroneni*	*Branta canadensis* (Anatidae)	USA	OQ780428	–	[30]
	*Circumvitellatrema momota*	*Momotus momota* (Momotidae)	France (captive-born in zoo)	JQ886067	–	[4]
	*Circumvitellatrema momota*	*Momotus momota * (Momotidae)	France (captive-born in zoo)	JQ886066	–	[4]
	*Cyclocoelum mutabile*	*Calidris canutus* (Scolopacidae), *Gallinula chloropus* (Rallidae)	Peru	AY222249	MH091807	[31,32]
	*Harrahium obscurum*	*Tringa ochropus* (Scolopacidae)	Russia	PP935228	PP934665	This study
	*Harrahium obscurum*	*Tringa nebularia*, *Tringa erythropus* (Scolopacidae)	Czech Republic	–	KU877888 KU877884 KU877890	[1]
	*Harrahium tringae*	*Tringa erythropus * (Scolopacidae)	Czech Republic	–	KU877889	[1]
	*Morishitium dollfusi*	*Pica pica* (Corvidae), *Turdus merula* (Turdidae)	Czech Republic, Italy	OR364989 OR364990 OR364991	OR364777 OR364782 MH800193 KU877887 KU877885	[1,6,7]
	*Morishitium dollfusi*	*Turdus merula* (Turdidae)	Russia	PP935227	PP934664	This study
	*Morishitium polonicum*	*Aplonis panayensis strigata* (Sturnidae)	Malaysia	LC520231	–	[33]
	*Morishitium polonicum malayense*	*Aplonis panayensis strigata* (Sturnidae)	Malaysia	–	LC517300 LC517302 LC517301	[33]
	*Neohaematotrephus arayae*	*Jacana spinosa* (Jacanidae)	Mexico	MH725787	–	[34]
	*Tracheophilus cymbius*	*Cairina moschata* (Anatidae)	China	MK327367	–	Not published
	*Typhlocoelum* sp.	*Anas platyrhynchos* (Anatidae)	USA	KT956960	–	[5]
	*Typhlocoelum cucumerinum*	*Cairina moschata* (Anatidae)	Brazil	–	MZ254997	[35]
Echinostomatidae	*Himasthla elongata*	*Littorina littorea* (Littorinidae)	Russia	MT987600	–	[36]

**Table 2 biology-13-00621-t002:** Measurements of *Harrahium obscurum* from the original material and redescriptions.

	This Study	Harrah [37], Dubois [38] on Original Material	Sitko et al. [1]	Lamothe-Argumedo, Orozco-Flores [39]
Host	*Tringa ochropus*	?	*Tringa erythropus*	*Tringa semipalmata*
Locality	Mordovia NR	USA	Czech Republic	Mexico
Body length	10.00–15.00 (11.96)	6.0–13.00	13.00–28.00 (20.71)	18.90–21.70
Body width	1.750–3.100 (2.392)	1.50–3.00	3.00–6.50 (4.80)	3.50–4.30
Pharynx width	0.183–0.362 (0.258)	0.115–0.264	0.183–0.350 (0.254)	0.270–0.372
Esophagus length	0.308–0.462 (0.374)	0.500–0.750	–	0.888 ^2^
Cirrus sac length	0.385–0.539 (0.442)	0.248–0.579	–	0.777 ^2^
Anterior testis length	0.323–0.492 (0.422)	0.462–0.827	0.486–1.143 (0.828)	1.110–2.180
Anterior testis width	0.400–0.539 (0.476)	0.380–0.827	0.457–1.222 (0.923)	1.440–2.500
Posterior testis length	0.354–0.585 (0.424)	0.480–1.000	0.514–1.457 (0.940)	1.010–1.819
Posterior testis width	0.385–0.662 (0.501)	0.300–0.877	0.514–1.341 (0.986)	1.300–1.980
Ovary length	0.262–0.339 (0.300)	0.275–0.463 ^1^	0.343–0.571 (0.482)	0.410–0.480
Ovary width	0.246–0.354 (0.286)		0.343–0.596 (0.483)	0.430–0.560
Intertesticular space length	0.658–1.785 (1.122)	0.830	–	1.888 ^2^
Post-testicular space length	0.400–0.754 (0.565)	0.415	–	0.444 ^2^
Egg length	0.109–0.139 (0.127)	0.138–0.162	0.133–0.157 (0.142)	0.144–0.161
Egg width	0.044–0.077 (0.069)	0.070–0.094	0.070–0.087 (0.082)	0.090

Note: Mean values are given in parentheses. ^1^—diameter. ^2^—estimated by Lopez-Jimenez et al. [30] from a published figure.

**Table 3 biology-13-00621-t003:** Measurements of *Morishitium dollfusi* from its original description and redescriptions.

	This Study	Timon-David [40]	Kirillova et al. [10]	Machalska [19]	Heneberg, Sitko [7]	Jaume-Ramis, Pinya [20]
Host	*Turdus merula*	*Pica pica*	*Turdus merula*, *Coccothraustes coccothraustes*	*Turdus philomelos*	*Turdus merula*, *Turdus philomelos*	*Turdus philomelos*
Locality	Zhiguli NR (Russia)	France	Smolny NP (Russia)	Poland	Czech Republic	Spain
Body length	8.00–11.75 (9.78)	15.50–20.00 (17.00)	6.250–8.254	7.138–13.109	6.850–21.500 (10.580)	12.00
Body width	1.90–2.75 (2.34)	–	1.815–2.650	–	2.000–4.100 (2.513)	2.50
Oral sucker width	0.169–0.285 (0.219)	0.350	0.169–0.266	0.208–0.323	184–368 (264) × 184–322 (246) ^2^	0.230
Pharynx width	0.221–0.277 (0.251)	0.250	0.193–0.266	0.208–0.323	193–285 (246) × 202–359 (261) ^2^	0.235
Esophagus length	0.185–0.339 (0.248)	0.250–0.425	–	–	0.230–0.644 (0.453)	–
Cirrus sac length	0.231–0.400 (0.299)	0.255	0.193–0.324	0.219–0.474	0.203–0.644 (0.368) × 116–0.230 (0.166) ^2^	0.365
Anterior testis length	0.385–0.769 (0.619)	0.910	0.370–1.000	0.498 ^1^	0.402–0.984 (0.687)	0.715
Anterior testis width	0.462–0.739 (0.589)	1.350			0.373–0.984 (0.592)	–
Posterior testis length	0.492–0.862 (0.679)	0.910			0.373–1.222 (0.820)	–
Posterior testis width	0.554–0.923 (0.695)	1.350			0.447–1.043 (0.701)	–
Ovary length	0.200–0.308 (0.263)	0.320–0.400	0.222–0.348	0.265 ^1^	0.184–0.507 (0.307)	0.280
Ovary width	0.262–0.354 (0.301)				0.230–0.745 (0.311)	–
Intertesticular space length	0.446–1.092 (0.759)	1.920	0.533–0.963	1.430	0.343–1.143 (0.773)	–
Post-testicular space length	0.677–1.062 (0.842)	1.590	0.444–0.954	1.260	0.571–1.714 (0.946)	–
Egg length	0.123–0.154 (0.135)	0.120–0.130	0.079–0.134	0.081–0.139	0.128–0.139 (0.137)	0.120
Egg width	0.064–0.079 (0.070)		0.043–0.071	0.046–0.069	0.070–0.081 (0.078)	0.060

Note: Mean values are given in parentheses. ^1^—diameter, ^2^—length and width are given.

## Data Availability

Data are contained within the article and available on request from the corresponding author.

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
