# Peer review of "The Morphological and Molecular Characterization of the Avian Trematodes Harrahium obscurum and Morishitium dollfusi (Digenea: Cyclocoelidae) from the Middle Volga Region (European Russia)"

_biology, 2024, doi:10.3390/biology13080621_

Round 1
Reviewer 1 Report
Comments and Suggestions for Authors
This is a useful early paper on the molecular and morphological systematics of cyclocoelids. It is too early to draw definite conclusions from the small amount of data available, but it seems clear that many of the standard morphological differentiating characters are of limited value. For example, the worm described in this paper as Harrahium obscurum (Leidy, 1887) based on the findings of Sitko et al. (2016), does not key to this genus using the characters in the review paper of Dronen & Blend (2015). The authors do not comment in any detail on what characters they still consider useful.
The paper is well written and illustrated and is worth publishing, but the authors could certainly draw attention to those morphological characters which their evidence suggests retain value as indicators of generic status.
The literature on this group is vast and as far as I can tell the authors have referred to most relevant papers. There are in the literature several articles describing the variation found in cyclocoelids which could have been consulted (see for example, the papers by Macko and colleagues cited in the Dronen & Blend (2015) paper).
Line 184. … not confluent merge posteriorly … This is contradictory, if the fields are not confluent, they do not merge, which appears to be the case.
Line 264. … extend above the intestinal bifurcation … - … extend anteriorly to the intestinal bifurcation …
Author Response
Comment 1: This is a useful early paper on the molecular and morphological systematics of cyclocoelids. It is too early to draw definite conclusions from the small amount of data available, but it seems clear that many of the standard morphological differentiating characters are of limited value. For example, the worm described in this paper as Harrahium obscurum (Leidy, 1887) based on the findings of Sitko et al. (2016), does not key to this genus using the characters in the review paper of Dronen & Blend (2015). The authors do not comment in any detail on what characters they still consider useful.
Response 1: Dear Reviewer, thank you for your interest and valuable comments on our work.
Comment 2: The paper is well written and illustrated and is worth publishing, but the authors could certainly draw attention to those morphological characters which their evidence suggests retain value as indicators of generic status.
Response 2: We agree with this comment. We have added relevant information to the Discussion. – Page 10 lines 274-275.
Comment 3: The literature on this group is vast and as far as I can tell the authors have referred to most relevant papers. There are in the literature several articles describing the variation found in cyclocoelids which could have been consulted (see for example, the papers by Macko and colleagues cited in the Dronen & Blend (2015) paper).
Response 3: Thank you for pointing this out. We agree with this comment. The works of Maсko et al. and some others were cited in our manuscript. – Page 10, lines 267-284.
Comment 4: Line 184. … not confluent merge posteriorly … This is contradictory, if the fields are not confluent, they do not merge, which appears to be the case.
Response 4: Agree. We have, accordingly, corrected sentence. – Page 7, line 184.
Comments 5: Line 264. … extend above the intestinal bifurcation … - … extend anteriorly to the intestinal bifurcation …
Response 5: Agree. We have, accordingly, corrected sentence. – Page 10, line 264.

Reviewer 2 Report
Comments and Suggestions for Authors
The manuscript examines the problematic issue of the taxonomic status of two species of avian trematodes, using morphological and molecular data. The authors presented data on morphological and morphometric characteristics of two cyclocoelid species - Harrahium obscurum from Tringa ochropus and Morishitium dollfusi from Turdus merula.
The authors are confident that the species identification of these trematodes was correct, but they revealed high variability of characters and some differences the morphometric characteristics of the studied specimens of both cyclocoelid species from other host species and regions. The authors concluded that some morphological features cannot be used in the morphological diagnosis of H. obscurum and M. dollfusi. Molecular data were partially consistent with previously published data. The phylogenetic analysis of partial 28S rRNA gene sequences revealed that Hyptiasminae and Cyclocoelinae are polyphyletic subfamilies.
This study makes a certain contribution both to the solution of problem of the identification of these two cyclocoelid species and to the understanding of phylogenetics of trematodes in general.
The originality of the text of the manuscript is more than 61%.
Since I am not a native English speaker, I did not rate the quality of English. But the English was clear and easy to read.
The text of the manuscript was carefully prepared, I only found isolated errors. Reference to figure 4 is missing in the text of the manuscript.
The manuscript can be accepted for publication after minor technical editing.
Author Response
Comments 1: The manuscript examines the problematic issue of the taxonomic status of two species of avian trematodes, using morphological and molecular data. The authors presented data on morphological and morphometric characteristics of two cyclocoelid species - Harrahium obscurum from Tringa ochropus and Morishitium dollfusi from Turdus merula.
The authors are confident that the species identification of these trematodes was correct, but they revealed high variability of characters and some differences the morphometric characteristics of the studied specimens of both cyclocoelid species from other host species and regions. The authors concluded that some morphological features cannot be used in the morphological diagnosis of H. obscurum and M. dollfusi. Molecular data were partially consistent with previously published data. The phylogenetic analysis of partial 28S rRNA gene sequences revealed that Hyptiasminae and Cyclocoelinae are polyphyletic subfamilies.
This study makes a certain contribution both to the solution of problem of the identification of these two cyclocoelid species and to the understanding of phylogenetics of trematodes in general.
The originality of the text of the manuscript is more than 61%.
Since I am not a native English speaker, I did not rate the quality of English. But the English was clear and easy to read.
Response 1: Dear Reviewer, thank you for your interest and kind words about our work.
Comments 2: The text of the manuscript was carefully prepared, I only found isolated errors. Reference to figure 4 is missing in the text of the manuscript.
Response 2: Agree. We have, accordingly, corrected text – Page 9, line 209.
The manuscript can be accepted for publication after minor technical editing.
